# Stroke in Young Adults in Spain: Epidemiology and Risk Factors by Age

**DOI:** 10.3390/jpm13050768

**Published:** 2023-04-29

**Authors:** Laura Amaya Pascasio, Marina Blanco Ruiz, Rodrigo Milán Pinilla, Juan Manuel García Torrecillas, Antonio Arjona Padillo, Cristina Del Toro Pérez, Patricia Martínez-Sánchez

**Affiliations:** 1Stroke Centre, Department of Neurology, Torrecárdenas University Hospital, 04009 Almería, Spain; 2Department of Emergency Medicine, Torrecárdenas University Hospital, 04009 Almería, Spain; 3Biomedical Research Unit, Torrecárdenas University Hospital, 04009 Almería, Spain; 4Instituto de Investigación Biomédica Ibs. Granada, 18012 Grranada, Spain; 5Centro de Investigación Biomédica en Red en Epidemiología y Salud Pública (CIBERESP), 28029 Madrid, Spain

**Keywords:** ischemic stroke, epidemiology, young, incidence, vascular risk factors

## Abstract

Introduction: Recent research has highlighted an increased incidence of ischemic stroke (IS) in young adults, along with a higher percentage of vascular risk factors at younger ages. This study aimed to estimate the in-hospital incidence of IS and associated comorbidities by sex and age group in Spain. Methods: A retrospective analysis of the Spain Nationwide Inpatient Sample database from 2016 to 2019 was conducted, which included adult patients with IS. In-hospital incidence and mortality rates were estimated, and a descriptive analysis of the main comorbidities was performed, stratified by sex and age groups. Results: A total of 186,487 patients were included, with a median age of 77 (IQR 66–85) years and 53.3% were male. Of these, 9162 (5%) were aged between 18 and 50 years. The estimated incidence of IS in adults younger than 50 years ranged from 11.9 to 13.5 per 100,000 inhabitants during the study period, with a higher incidence in men. The overall in-hospital mortality was 12.6%. Young adults with IS had a higher prevalence of most vascular risk factors compared to the general Spanish population, with a specific distribution according to sex and age. Conclusions: This study provides estimates of the incidence of IS and the prevalence of vascular risk factors and comorbidities associated with IS in Spain, stratified by sex and age, using a national registry of hospital admissions. These findings should be considered in terms of both primary and secondary prevention strategies.

## 1. Introduction

Stroke is a significant health, social, and economic problem representing the second leading cause of death and the first cause of adult disability in Europe [1]. The population based IBERICTUS epidemiological study estimated the incidence of ischemic stroke (IS) in Spain at 118 cases per 100,000 inhabitants, with the majority of cases occurring in those over 65 years of age, with a predominance of men [2]. This incidence is somewhat higher than that recorded in other countries such as the United States, England, Germany, Poland, and France [2]. Although it is estimated that approximately 75% of strokes affect patients over 65 years of age and around 5% of all strokes affect people between 18 and 44 years of age, studies carried out over the last decade warn of an increasing incidence of IS in adults under 55 years of age [3,4,5,6,7,8,9]. Possible explanations for this rising incidence include better detection and awareness of the disease, as well as an increased prevalence of modifiable traditional risk factors [5,10,11,12]. A population-based study conducted in England from 2002 to 2018 reported a significant increase in stroke incidence in individuals younger than 55 years, with a higher prevalence of vascular risk factors in younger stroke patients compared to the general population [13]. In addition, there is evidence of a considerable percentage of undiagnosed vascular risk factors in people under 55 years of age who suffer a first cerebrovascular event [14].

This increase in the incidence of IS in younger people has devastating socio-economic consequences and highlights the need for public health programs to combat traditional risk factors, as well as identify new factors contributing to an increased risk of stroke at an earlier age [15]. However, there is a lack of contemporary information on the prevalence of vascular risk factors and other comorbidities among young stroke patients in Southern Europe. Therefore, this study aims to estimate the in-hospital incidence of IS in young adults in Spain and the main comorbidities by age group and sex using the Spanish Nationwide Inpatient Database.

## 2. Materials and Methods

### 2.1. Type of Study

A retrospective cohort study was conducted using analytical observation of all hospital admissions for a primary diagnosis of stroke in the Nationwide Inpatient Database of Spain, which has a population of approximately 47 million, during a 4-year period from 2016 to 2019. The study was approved by the Clinical Ethics Committee of the Province of Almeria, Torrecárdenas University Hospital, Andalusian Health Service, Ministry of Health, Andalusia (Spain).

### 2.2. Patients

This study used the Spanish Minimum Basic Hospital Discharge Dataset, which was provided by the Ministry of Health, Consumerism, and Social Policies, as the source of information. Diagnostic and procedural coding was performed according to the Tenth Revision of the International Classification of Diseases (ICD10). The study included patients who were discharged with a primary diagnosis of IS, identified by ICD10 codes I63.0, I63.1, I63.2, I63.4, I63.5, I63.7, I63.8, I63.9, and subgroups. The code I63.3 was excluded since it refers to IS secondary to cerebral venous sinus thrombosis. Cases of transient ischemic attack were not included. The presence of vascular risk factors or other relevant comorbidities was obtained from the secondary diagnoses listed for each case, with the relevant ICD10 codes provided in Appendix A.

Duplicate entries for patients admitted to different institutions were identified by cross-referencing anonymized identification codes and dates of birth, and periods of hospitalization were traced to merge duplicate entries into a single record. The number of patients diagnosed with IS, the in-hospital mortality rate, and the main comorbidities at discharge were analyzed by sex and age group (≤50 or >50).

### 2.3. Study Variables

Incidence was calculated as the number of cases of hospitalization per year divided by the population at risk (according to the age range of the cases) registered in the national statistics institute at mid-year. Overall and age-specific mortality rates were calculated by analyzing the percentage of deaths during admission.

Comorbidities were categorized into vascular risk factors, including arterial hypertension, diabetes mellitus, dyslipidemia, obesity, and tobacco use; toxic habits such as alcohol or illicit drug use; and other common comorbidities such as heart, renal or pulmonary disease, and cancer. The presence of infrequent but potentially etiological conditions for IS such as fibromuscular dysplasia, endocarditis, arterial dissection, or thrombophilia was also analyzed. Finally, the presence of complications during admission such as cerebral herniation or infections was recorded.

### 2.4. Data Analysis

To analyze the presence of comorbidities based on age group and sex, the SPSS 27.0 (SPSS Inc., Chicago, IL, USA) program for macOS was used for the statistical analyses. Continuous variables were presented as mean (standard deviation (SD)) or median (interquartile range (IQR)), while categorical variables were presented as percentages. The chi-squared test was employed for dichotomous variables in univariate analysis, and the *t*-test was used for continuous variables with a normal distribution. A significance level of *p* < 0.05 was considered statistically significant. Patients were categorized into two groups based on age (≤50 or >50) for univariate and multivariate logistic regression analysis. Furthermore, changes in the prevalence of vascular risk factors were analyzed by sex and 5-year age intervals.

To estimate the incidence of IS by sex and age group, the number of new hospitalized cases each year was calculated and divided by the population at risk, which was registered in the Spanish mid-term census, with data obtained from the national statistics institute. To stratify the estimated incidence of IS by the presence of vascular risk factors in the general population, following the methodology previously reported by Li et al. [13], data from the Health Survey for Spain conducted in 2014 was utilized. This survey included 22,321 participants aged 18 years or older, representing the Spanish population. The analysis was based on the cases of IS occurring in 2016 and the Spanish population at risk during this year, which was the closest year to the analyzed health survey.

## 3. Results

A cohort of 186,487 hospital discharges with a confirmed diagnosis of IS was analyzed after eliminating duplicate cases. The man age of the cohort was 74.7 years (SD 13.4), with males constituting 53.3% of the total. Of these, 9162 cases (5%) were individuals between 18 and 50 years of age. The estimated incidence of IS by sex and age group during the analyzed period is presented in Table 1. The overall in-hospital mortality rate was 12.6%, with a range of 1.8% for younger age groups to 20.2% for individuals aged over 80 years (Appendix A).

Table 2 presents the primary demographic and clinical characteristics of the study cohort stratified by age group. The prevalence of classical vascular risk factors, such as hypertension or dyslipidemia, was lower in the younger age group; however, it increased progressively and reached up to 30%. In addition, smoking had a prevalence of up to 50% in this group. A gradual increase in the prevalence of these risk factors was observed, particularly after the age of 30. Figure 1 displays the changes in the absolute prevalence of vascular risk factors and stroke-related comorbidities across 5-year age ranges, with more detailed information available in Appendix A.

In the age group of 18–50 years, female patients were significantly younger and constituted the majority in the age subgroup below 30 years. Furthermore, there were notable differences in the distribution of comorbidities related to IS with respect to age and sex. The prevalence of traditional vascular risk factors showed a greater difference by sex in the younger age group. Men in this group had a significantly higher percentage of high blood pressure (33% vs. 24.2%), diabetes mellitus (12% vs. 7.8%), and dyslipidemia (30.2% vs. 19.6%) compared to women. Likewise, women had a significantly lower prevalence of illicit drug use but a higher prevalence of obesity and atrial fibrillation, among other conditions. The prevalence of atrial fibrillation or heart failure was higher in men in the younger age group, but it was reversed when analyzing only the group of older people with IS. Additional information regarding the distribution of comorbidities by sex and age group is available in Appendix A.

Compared to the general population sample included in the 2014 health survey (see Appendix A), the analyzed cohort exhibited a higher prevalence of smoking, hypertension, and diabetes mellitus, while the prevalence of obesity and dyslipidemia was slightly lower. Figure 2 displays the stratified estimated incidence of IS in adults aged 18 to 50, based on the prevalence of vascular risk factors in the general population of the same age group, as reported in the 2014 health survey data.

## 4. Discussion

The present study evaluated the incidence of IS in Spain according to sex and age groups using data from a national hospital admission registry. The results indicated that the estimated incidence of IS was consistently higher among men across all age groups, compared to women. Furthermore, the studied cohort exhibited a greater prevalence of most vascular risk factors when compared to the overall Spanish population. These risk factors were distributed differently across genders and age groups, highlighting the need for careful consideration when planning primary and secondary prevention strategies.

The estimated incidence of IS in the analyzed cohort was found to be slightly higher than that reported in the IBERICTUS study [2]. However, for the age group below 50, the incidence was similar to that reported in previous studies conducted in Spain, such as the one by Tejada et al. [16], which followed a similar methodology and was carried out in a northern region of Spain. Other epidemiological studies conducted in Europe over the past decade have reported a range of incidence rates from 7.4 to 45/100,000 [10,17,18]. The significant variability in incidence rates could be attributed to methodological differences, such as the use of population-based versus hospital records, as in the present work. Additionally, the age ranges of the analyzed population varied across different cohorts, with stroke in young patients typically defined as those between 15 and 55 years old, with the upper limit usually set at 45 or 50 years old.

A significant rise in the incidence of IS among young patients has been observed in recent epidemiological studies conducted in both Europe and North America [5,7,8,9,19]. This trend was also evident during the period analyzed in this research. While this increase could be attributed to improved diagnosis due to more accessible and advanced imaging techniques and greater public awareness of cerebrovascular disease, it has also been linked to the growing prevalence of modifiable risk factors in these populations, including classic vascular risk factors, obesity, and increased consumption of illicit drugs [13,20,21]. A population-based registry from Dijon reported a similar increase in stroke incidence over a 27-year period, along with a rise in IS among individuals under 55 years of age, consistent across both sexes [7]. The most prevalent modifiable risk factor in this registry was smoking, which was also the case in individuals under 50 years in our cohort. Other classic risk factors, such as arterial hypertension, diabetes mellitus, and dyslipidemia, also displayed a similar distribution to that observed in our study.

Although IS in young adults is often associated with uncommon causes such as arterial dissection or thrombophilia, which were more prevalent in this age group in our cohort, previous studies have shown that classical vascular risk factors have a comparable prevalence in this stroke group to that of older patients [11]. Studies conducted in European countries that included young adults with IS reported smoking rates of 50–55%, physical inactivity at 48.2%, arterial hypertension between 30–46%, diabetes mellitus at 6–11%, dyslipidemia at 34.9–45%, and obesity at 22.3%, which were slightly higher than the rates observed in our study [22,23]. Differences among studies may be due to variations in lifestyles and health policies between countries, as well as methodological heterogeneity since some comorbidities may be underdiagnosed in retrospective hospital-based databases. In all these cohorts, the prevalence of these risk factors increased significantly with age, particularly for arterial hypertension, a trend that we also observed in our work [10,16,22,23]. When evaluating the progression of vascular risk factors by age and sex, there was an accumulation of conditions such as hypertension, dyslipidemia, or diabetes, particularly in males, showing an exponential increase after 30 years of age (Figure 1).

Young adults who experience IS have been found to have higher rates of vascular risk factors compared to the general population [13,24], a trend also observed in this study when compared to the percentage of these risk factors in the same age group from a population-based survey representative of the Spanish population. Mitchell et al. conducted a case–control study and found that obesity was significantly associated with an increased risk of IS in younger adults aged 15 to 49 years [20]. Abdominal obesity has also been identified as an independent risk factor for cryptogenic IS in young adults [25]. Moreover, the number of well-documented vascular risk factors has been independently associated with higher long-term mortality in this population [26].

Another factor that is associated with the pathogenesis of stroke, especially in younger populations, is the consumption of toxic substances such as alcohol, tobacco, or illicit drugs [21]. In this study, the percentage of toxic substance use was significantly higher in younger men. Tobacco is a well-known risk factor for stroke, with a relative risk of up to 2.9 in people younger than 55 years [26,27]. Similarly, many illicit drugs, such as cannabis and cocaine are also associated with the pathogenesis of stroke [28,29]. De Los Rios et al. reported that one in five young adults with stroke abused illegal drugs [21], which underscores the importance of identifying substance abuse, especially in younger populations. In our study, cocaine was the most prevalent drug.

Arterial dissection, hereditary thrombophilia, patent foramen ovale, neoplasms, or systemic inflammatory diseases are non-modifiable risk factors associated with IS in young populations [11]. The prevalence of arterial dissection and hereditary thrombophilia varies in different studies, ranging from approximately 3–12% and 6–24%, respectively [10,30,31]. In our cohort, we observed similar rates of thrombophilia in the youngest age group and a lower frequency of arterial dissection. Malignancy is an increasingly recognized risk factor for stroke at a younger age, with a prevalence of 1.6% observed in our study, being higher in younger women and increasing with age. The Teenage and Young Cancer Survivor study, a large cohort study of patients aged 15–39 years, showed a 50% higher incidence of IS following malignancy [29]. Previous data from 1002 young adults with stroke in Finland showed that up to 8% of patients were found to have a malignancy, of which 4% were diagnosed before the stroke [30].

The present study has some limitations that may affect the generalizability of the results. Firstly, the retrospective design of the study may limit the availability of information on stroke etiology and location. Secondly, the study is based on a hospital database, which may not include all cases of IS occurring in Spain. However, hospitalization is common for patients with IS, especially in the younger age group, as it is a severe and potentially life-threatening condition. Previous analyses have reported hospitalization rates ranging from 70 to 90%, depending on various factors such as age and stroke severity [32]. Additionally, comorbidities may be underrepresented, as they may not be coded as a diagnosis at discharge. Lastly, the cohort only includes individuals residing in Spain, and the socio-economic status of the included cases is not available, which may limit the generalization of the findings to other regions of the world.

Despite this, the study provides valuable information on the prevalence and distribution of IS risk factors among young adults in Spain, which is consistent with previous research. Moreover, it draws attention to a growing concern: the increasing incidence of IS and vascular comorbidities in individuals under 50 years of age. Such findings have the potential to shape health and education policies, including the prospect of introducing health education programs from an early age, for instance, in schools or as part of health education initiatives [33].

## 5. Conclusions

The occurrence of IS among young adults is not uncommon in Spain, as evidenced by the considerable number of cases observed in this large cohort. The prevalence of modifiable risk factors was found to be high, particularly in older patients and males. These findings highlight the importance of implementing proactive primary and secondary prevention strategies that specifically target modifiable vascular risk factors in young populations using gender-specific approaches.

## Figures and Tables

**Figure 1 jpm-13-00768-f001:**
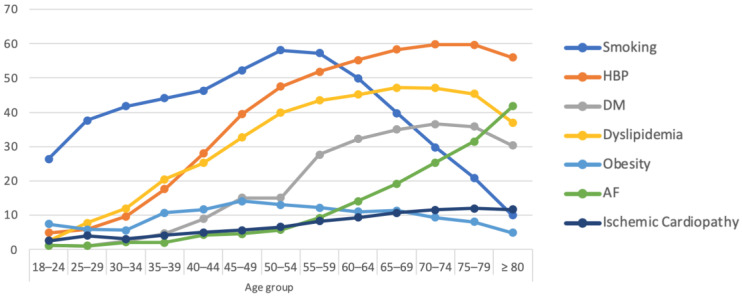
Progression of the prevalence (%) of risk factors for stroke by age group. HBP: High Blood Pressure; DM: Diabetes Mellitus; AF: Atrial Fibrillation.

**Figure 2 jpm-13-00768-f002:**
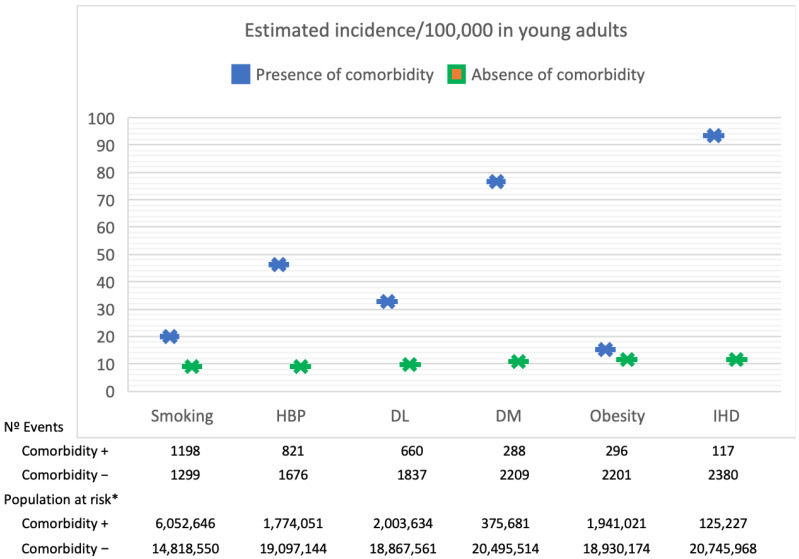
Stratified estimated incidence of ischemic stroke in adults aged between 18 and 50 years, categorized by the prevalence of vascular risk factors in the general population of the same age, * as per the data from the 2014 health survey. HBP: High Blood Pressure; DL: Dyslipidemia; DM: Diabetes Mellitus; IHD: Ischemic heart disease.

**Table 1 jpm-13-00768-t001:** Estimated incidence of hospitalized patients with ischemic stroke per 100,000 inhabitants categorized by sex and age group during the period from 2016 to 2019.

Age Group	≥18 (All)			18–50 (Young)			>50 (Old)		
Year	All	Men	Women	All	Men	Women	All	Men	Women
2016	111.6	122.2	101.6	11.9	15.4	8.4	232.2	263.6	205.4
2017	118.9	130.6	107.9	12.4	15.8	8.9	244.2	278.2	214.9
2018	124.9	138.1	112.6	12.8	16.5	9	253.5	290	222.1
2019	130.4	143	118.5	13.5	12.3	9.7	261.5	296.7	231.2

**Table 2 jpm-13-00768-t002:** Demographic and clinical characteristics of the included patients by age group. Bivariate and multivariate logistic regression analysis.

Variable	All *n* = 186,487	18–50 Years *n* = 9162 (5%)	>50 Years *n* = 1,177,325 (95%)	*p*	Odds Ratio	*p*
		Bivariate Analysis			Multivariate Analysis	
Demographic data, vascular risk factors and comorbidities						
Age, (mean, SD), years	74.7 (13.4)	43.5 (6.3)	76.6 (11.2)	<0.001		
Male. *n* (%)	99,408 (53.3%)	5874 (64.1%)	93,534 (52.7%)	<0.001	1.22 (1.16;1.28)	<0.001
Arterial hypertension. *n* (%)	102,963 (55.2%)	2736 (29.9%)	100,227 (56.5%)	<0.001	2.588 (2.468;2.713)	<0.001
Diabetes mellitus. *n* (%)	57,557 (30.9%)	964 (10.5%)	56,593 (31.9%)	<0.001	2.481 (2.323;2.649)	<0.001
Dyslipidemia. *n* (%)	76,136 (40.8%)	2417 (26.4%)	73,719 (41.6%)	<0.001	1.535 (1.461;1.613)	<0.001
Smoking. *n* (%)	48,457 (26%)	4408 (48%)	44,049 (24.8%)	<0.001	0.446 (0.4250.468)	<0.001
Cocaine, *n* (%)	865 (0.5%)	481 (5.2%)	384 (0.2%)	<0.001	0.100 (0.084;0.118)	<0.001
Cannabis, *n* (%)	772 (0.4%)	398 (4.3%)	374 (0.2%)	<0.001	0.216 (0.180;0.259)	<0.001
Alcohol, *n* (%)	11,773 (6.3%)	864 (9.4%)	10,909 (6.2%)	<0.001	1.359 (1.253;1.474)	<0.001
Obesity, *n* (%)	14,860 (8%)	1107 (12.1%)	13,753 (7.8%)	<0.001	0.380 (0.354;0.408)	<0.001
Migraine, *n* (%)	1626 (0.8%)	641 (6.5%)	985 (0.6%)	<0.001	0.164 (0.145;0.186)	<0.001
Hypothyroidism, *n* (%)	10,231 (5.5%)	329 (3.6%)	9902 (5.6%)	<0.001	1.216 (1.082;1.366)	0.003
Hyperthyroidism, *n* (%)	1516 (0.8%)	63 (0.7%)	1453 (0.8%)	0.17	0.819 (0.631;1.064)	0.107
Cancer diagnosis, *n* (%)	8431 (4.5%)	143 (1.6%)	8288 (4.7%)	<0.001	2.874 (2.446;3.378)	0.358
Ischemic heart disease, *n* (%)	19,887 (10.7%)	458 (5%)	19,429 (11%)	<0.001	1.770 (1.604;1.954)	<0.001
Atrial fibrillation, *n* (%)	54,425 (29.2%)	346 (3.8%)	54,079 (30.5%)	<0.001	6.777 (6.116;7.510)	<0.001
Non-congenital cardiopathy (all), *n* (%)	3918 (2.1%)	240 (2.6%)	3678 (2.1%)	<0.001	0.459 (0.379;0.555)	<0.001
Dilated cardiomyopathy, *n* (%)	1883 (1%)	114 (1.2%)	1769 (1%)	0.02	0.899 (0.685;1.179)	0.402
Heart failure, *n* (%)	10,824 (5.8%)	103 (1.1%)	10,721 (6%)	<0.001	2.095 (1.735;2.529)	<0.001
Valvulopathy (all), *n* (%)	10,959 (5.9%)	185 (2%)	10,774 (6.1%)	<0.001		
Valvulopathy (non-rheumatic), *n* (%)	6220 (3.3%)	87 (0.9%)	6133 (3.5%)	<0.001	2.626 (2.119;3.255)	<0.001
Valvulopathy (rheumatic), *n* (%)	5146 (2.8%)	104 (1.1%)	5042 (2.8%)	<0.001	1.426 (1.167;1.743)	0.001
PFO, *n* (%)	2224 (1.2%)	944 (10.3%)	1280 (0.7%)	<0.001	0.140 (0.127;0.155)	<0.001
Chronic renal failure, *n* (%)	20,221 (10.8%)	208 (2.3%)	20,013 (11.3%)	<0.001	3.955 (3.454;4.530)	<0.001
COPD, *n* (%)	10,555 (5.7%)	55 (0.6%)	10,500 (5.9%)	<0.001	8.563 (6.815:10.758)	<0.001
OSAH, *n* (%)	7962 (4.3%)	366 (4%)	1596 (4.3%)	0.19	1.064 (0.952;1.191)	0.361
Intracranial atherosclerosis, *n* (%)	7320 (3.9%)	155 (1.7%)	7165 (4%)	<0.001	2.398 (2.051;2.806)	<0.001
Unusual causes of stroke						
Fibromuscular dysplasia, *n* (%)	36 (0%)	14 (0.2%)	22 (0%)	<0.001	0.262 (0.116;0.591)	0.002
Arterial dissection (all), *n* (%)	988 (0.5%)	401 (4.4%)	587 (0.3%)	<0.001		
Arterial dissection (intracranial), *n* (%)	77 (0%)	32 (0.3%)	45 (0%)	<0.001	0.127 (0.075;0.217)	<0.001
Arterial dissection (carotid), *n* (%)	619 (0.3%)	222 (2.4%)	397 (0.2%	<0.001	0.183 (0.152;0.221)	<0.001
Arterial dissection (vertebral), *n* (%)	302 (0.2%)	151 (1.6%)	151 (0.1%)	<0.001	0.094 (0.072;0.122)	<0.001
Thrombophilia, *n* (%)	1111 (0.6%)	261 (2.8%)	850 (0.5%)	<0.001	0.262 (0.210;0.326)	<0.001
Antiphospholipid syndrome, *n* (%)	387 (0.2%)	132 (1.3%)	255 (0.1%)	<0.001	0.721 (0.520;1.000)	0.046
Systemic vasculitis (all), *n* (%)	358 (0.2%)	25 (0.3%)	333 (0.2%)	<0.001	0.701 (0.439;1.121)	0.118
Moyamoya, *n* (%)	32 (0%)	13 (0.1%)	19 (0%)	<0.001	0.070 (0.032;0.156)	<0.001
Sick cell disease, *n* (%)	17 (0%)	8 (0.1%)	9 (0%)	<0.001	0.098 (0.033;0.294)	<0.001
Fabry disease, *n* (%)	14 (0%)	7 (0.1%)	7 (0%)	<0.001	0.104 (0.032;0.337)	<0.001
Endocarditis, *n* (%)	204 (0.1%)	166 (0.2%)	188 (0.1%)	0.053	0.421 (0.238;0.747)	0.007
Cardiac prothesis infection, *n* (%)	335 (0.2%)	24 (0.3%)	311 (0.2%)	0.056	0.691 (0.428;1.115)	0.129
In-hospital complications						
Brain herniation, *n* (%)	1053 (0.6%)	87 (0.9%)	966 (0.5%)	<0.001	0.318 (0.247;0.409)	<0.001
Seizures, *n* (%)	1187 (0.6%)	46 (0.5%)	1141 (0.6%)	0.097	0.824 (0.607;1.120)	0.421
Acute coronary syndrome, *n* (%)	779 (0.4%)	26 (0.3%)	753 (0.4%)	0.04	0.697 (0.482;1.009)	0.229
Urinary tract infection, *n* (%)	13,598 (7.3%)	207 (2.3%)	13,391 (7.6%)	<0.001	2.192 (1.912;2.513)	<0.001
Acute renal failure, *n* (%)	6734 (3.6%)	95 (1%)	6639 (3.7%)	<0.001	1.473 (1.200;1.807)	<0.001
Broncoaspiration, *n* (%)	7693 (4.1%)	84 (0.9%)	7609 (4.3%)	<0.001	2.143 (1.732;2.653)	<0.001
Pneumonia, *n* (%)	2499 (1.3%)	34 (0.4%)	2465 (1.4%)	<0.001	1.916 (1.375;2.668)	<0.001
Outcome						
In-hospital mortality, *n* (%)	23,563 (12.6%)	253 (2.8%)	23,310 (13.1%)	<0.001	2.917 (2.576;3.303)	<0.001

YO: Years old; PFO: Patent foramen oval, COPD: Chronic obstructive pulmonary disease; OSAH: Obstructive sleep apnea-hypopnea syndrome.

## Data Availability

Data of this study are available under considerable request.

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
