# Peer review of "Stroke in Young Adults in Spain: Epidemiology and Risk Factors by Age"

_jpm, 2023, doi:10.3390/jpm13050768_

Round 1
Reviewer 1 Report
Dear Editor and authors,
Thank You for the opportunity to review this interesting paper. The stroke is one of the leading reason of mortality in Europe, and leading course of long term mortality. The correction of risk factors results in decrease the incidence and burden of stroke. Perhaps. the part of young stroke patients increased among of stroke patients and knowledge about risk profile in tis group is important. So, the aim of this paper in actual.
The paper is written in good style, methods and study population described clearly. The results presented clearly, table and figures are appropriate. The conclusion is addressed to results. However, the paper has few important concerns, .
1. In line 68 authors note, that the patients with ICD -10 code I60-69 were included in this study. However, the ICD -10 codes I65 - I69 do not include stroke patients , therefore inclusion of these patients in this studies could significant change results. Secondly, the risk factors of intracerebral hemorrhage (I61-62) and ischemic stroke (I63) are different and should be analyzed separately. The same is about subarachnoid hemorrhage. Please do separate analysis or discuss it in the discussion section.
2. Was the same study performed early? If yes it would be interesting to compare the results and comment how profile of risk factors changed during the period
Author Response
- In line 68 authors note, that the patients with ICD -10 code I60-69 were included in this study. However, the ICD -10 codes I65 - I69 do not include stroke patients, therefore inclusion of these patients in this studies could significant change results. Secondly, the risk factors of intracerebral hemorrhage (I61-62) and ischemic stroke (I63) are different and should be analyzed separately. The same is about subarachnoid hemorrhage. Please do separate analysis or discuss it in the discussion section.
As the reviewer rightly points out, there has been a mistake in the description of the codes used, since although we requested information on codes I63 - I69, not all of them were used in the current study but only those referring to ischemic stroke: I63 and subgroups (I63.0 to I63.9), I63.6 referring to pyogenic venous thrombosis was also excluded. This error has been corrected in the manuscript.
- Was the same study performed early? If yes it would be interesting to compare the results and comment how profile of risk factors changed during the period
Unfortunately, data from previous periods is not available.
Reviewer 2 Report
Introduction Good
Method
How do you get the information about comorbidities? If from the database, please describe the ICD10 Code of the comorbidities inn detail.
Statistics; p 2-tailed?
Results
Is it really a non-normal distribution? Although in many cases, such surveys are normally distributed.
Please also do a multivariate analysis and further clarify the difference between 50+ and less than....please give Odds and show and discuss the difference between 50< and 50< where younger people fall into IS.
Discussion very good.
Describe the importance of raising awareness from an early stage.
https://www.researchgate.net/publication/369761250
- PMID: 25523059
Author Response
Method section
How do you get the information about comorbidities? If from the database, please describe the ICD10 Code of the comorbidities detail.
Information about comorbidities was obtained from the secondary diagnosed coded in the database for each case. The ICD10 codes used for each comorbidities have been added to the supplementary material.
Statistics; p 2-tailed
One-tailed tests have been applied.
Results
Is it really a non-normal distribution? Although in many cases, such surveys are normally distributed.
The reviewer is right, the distribution is normal and parametric tests and mean (SD) have been used.
Please also do a multivariate analysis and further clarify the difference between 50+ and less than....please give Odds and show and discuss the difference between 50< and 50< where younger people fall into IS.
As suggested by the reviewer, a multi-variate analysis including the relative weight (odds ratio) of each risk factor/comorbidity has been added to table 2.
Discussion very good.
Describe the importance of raising awareness from an early stage.
https://www.researchgate.net/publication/369761250; PMID: 25523059
A reflection on the need for early education (for instance in schools) on stroke and its comorbidities has been added to the discussion.
Reviewer 3 Report
This is an excellent paper dealing with the risk factors of stroke in young adults in Spain.
I think this study is very significant, because there is a background that stroke is increasing among young adults in Spain.
I think that the results are not well discussed and conclusions are not meaningful in manuscript. I would like to point out two things to fix.
1. Multivariate analysis should be used to examine results and odds ratios should be provided.
2. Since there is no consideration of the social background in Spain, it has lost its significance although this data is extremely important.
English of the manuscript has to be checked by a person familiar with sociologist too.
Author Response
- Multivariate analysis should be used to examine results and odds ratios should be provided.
As suggested by the reviewer, a multi-variate analysis including the relative weight (odds ratio) of each risk factor/comorbidity has been added to table 2.
- Since there is no consideration of the social background in Spain, it has lost its significance although this data is extremely important.
As the reviewer says, the absence of information on socio-economic information on the included stroke cases is a limitation and has been added to the manuscript.
English of the manuscript has to be checked by a person familiar with sociologist too.
Following this suggestion, English has been revised and improved
Round 2
Reviewer 1 Report
Dear authors ands editor,
Thank You for interesting manuscript. The manuscript is improved significantly, and the study design now is much better and clearer.
However, please double check the ICD code in Patient section: you noted, that I63.3 code was excluded (it refers secondary stroke due to venous thrombosis), but really I63.3 refers ischemic stroke due to cerebral arteries thrombosis. I63.6 - the code of IS due to cerebral venous thrombosis.
There is no code I63.7 at all.
Reviewer 3 Report
- The manuscript has been revised well. I think this manuscript will be acceptable after some corrections have been done.